# GRAPH SIMILARITIES AND DUAL APPROACH FOR SEQUENTIAL TEXT-TO-IMAGE RETRIEVAL

## ABSTRACT

Sequential text-to-image retrieval, a.k.a. Story-to-images task, requires semantic alignment with a given story and maintaining global coherence in drawn image sequence simultaneously. Most of the previous works have only focused on modeling how to follow the content of a given story faithfully. This kind of overfitting tendency hinders matching structural similarity between images, causing an inconsistency in global visual information such as backgrounds. To handle this imbalanced problem, we propose a novel image sequence retrieval framework that utilizes scene graph similarities of the images and a dual learning scheme. Scene graph describes high-level information of visual groundings and adjacency relations of the key entities in a visual scene. In our proposed retriever, the graph encoding head learns to maximize embedding similarities among sampled images, giving a strong signal that forces the retriever to also consider morphological relevance with previously sampled images. We set a video captioning as a dual learning task that reconstructs the input story from the sampled image sequence. This inverse mapping gives informative feedback for our proposed retrieval system to maintain global contextual information of a given story. We also suggest a new contextual sentence encoding architecture to embed a sentence in consideration of the surrounding context. Through extensive experiments, Our proposed framework shows better qualitative and quantitative performance with Visual Storytelling benchmark compared to conventional story-to-image models.

## 1 INTRODUCTION

Visual content and textual description are in synergytic relations, have advantageous on delivering information and being a proper expression means in real world communications. In this sense, successful cross-modal learning requires neural networks to deeply comprehend semantic relations between corresponding visual concepts and their textual descriptions. Examples include learning joint representation(Li et al., 2015; Ma et al., 2021; Pan et al., 2013), text generation from visual depiction(Farhadi et al., 2010; Kulkarni et al., 2013; Socher et al., 2014), and image or video retrieval from text queries(Kim et al., 2015; Zitnick et al., 2013). Such multi-modal tasks build on common embedding space where semantically associated visual and text embeddings are jointly mapped into similar location. Hence, the key of multi-modal learning is understanding semantic relationship between distinct modality representations.

In this sense, image-text retrieval have been widely explored as a one of the core tasks in the multi-modal learning. The performance of such retrieval system has been measured by examining how the retrieved samples are semantically aligned with given query. In addition, most of them have been trained on singleton image-text pair dataset(e.g., COCO(Lin et al., 2015)). These requirement and condition enforces the existing retrieval system to output the sample which is the most semantically fitted to the query. In other words, the performance of the system is dominantly determined by the level of overfitting tendency of the output. The more output sample is relational to the input, the more performance increases. This comes with potential arguments in sequential cross-modal learning since it only cares *inter* relationship with certain modality input at given time step, while neglecting the *intra* relations of output modality. Consequently, when we qualitatively analyze output samples, we can observe some incoherencies between output samples.

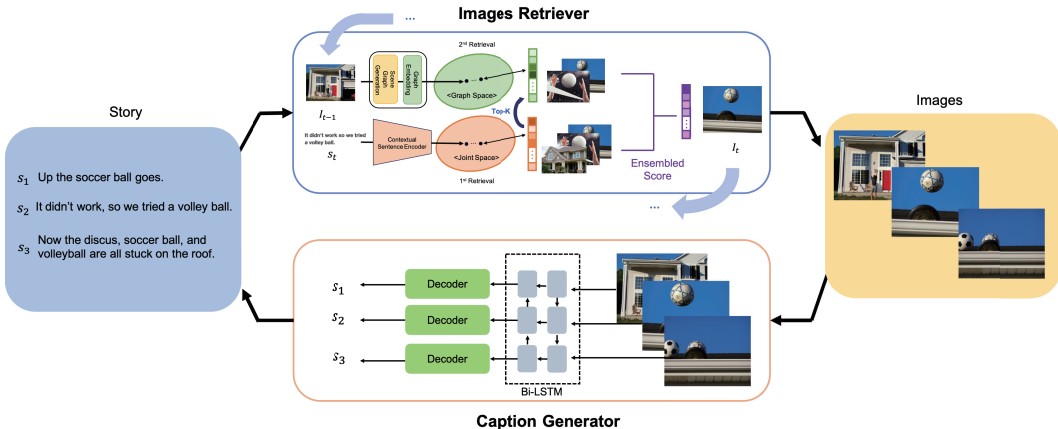

Figure 1: An Overall illustration of GD framework

In sequential text-to-image, many of conventional approaches attempt to maximize the visual coherence with given set of descriptions. Though they largely improved modeling a cross-modal coherency between text and image, many of them neglect scenery coherence among images. In other words, since the system focuses on reflecting objects, optical discrepancies frequently occur between neighboring images. For example, peripheral visual cues like background objects, weather, and location changes as the story proceeds. We view this problem as an optimization problem whose goal is alleviating the one-sided overfitting to a certain modality. To address this, we have to consider an additional method which can teach the retriever to also count similarity in the other modality.

To overcome the suggested issue, we should handle two main challenges. First, we also have to measure how much are two successive landscapes similar, that is, we need an organized representation that includes spatial information of each photo. Second, even if an image sequence was chosen considering scenery similarity, the curated image sequence must be still capable of narrate given input story. In this paper, we address aforementioned challenges with scene graph structure and dual learning framework.

Scene graph(Johnson et al., 2015) is a graph representation that includes abstract summaries of the objects and their relationships within an image. The objects are represented as nodes, and the relations between them is usually represented as bidirectional edges. Due to it's canonicalized description, scene graphs are very effective structured representation to easily figuring out global information of both images and language. We generate scene graph of every image in training set to provide extra spatial information for the retriever. Hence, we need to also develop a encoding module to process scene graphs. In this paper, we attach GCN(graph convolutional network)(Kipf & Welling, 2016) as a scene graph encoding head of the retriever which extracts compressed representation from nodes and edges from given scene graph. By drawing the scene graphs of retrieved image sequence and passing to the following encoding head, model can compute the extent of their similarities in graph embedding level.

Dealing with second challenge, we utilized dual learning framework which first introduced by Xia et al. (2016). In dual learning, two agents are involved where one agent solves primal task and the other solves dual task. For example, Xia et al. (2016) set English-to-French translation as a primal, and French-to-English as a dual. This cyclic loop gives each agent to learn error feedback from each other. We set story-to-images as a primal task, and images-to-story, which can be regarded as video captioning, be a dual task. The main purpose of leveraging dual framework is providing informative error signal from dual task to the primal retriever system. The signal will teach the retriver to pick well-curated photos which become the input for the dual agent. In overall, dual task contributes to enhance the qualitative confidence for an output of the retriver.

Figure 1 depicts our aforementioned approaches in a single architecture. We experiment our approaches to the most popular visual storytelling benchmark: VIST(Huang et al., 2016). Specific configuration of VIST will be explained in section 4.

In a nutshell, out key contributions are as follows:

1. We propose end-to-end sequential text-to-image retriever which achieves semantic coherency in both inter and intra modalities, which resolves overfitting on single modality

2. We explore visual storytelling, which is non-trivial task that requires balance in semantic dependencies between distinct modalities.

## 2 RELATED WORKS

### 2.1 IMAGE RETRIEVAL

Much cross-modal retrieval research has dealt with learning a latent space that jointly embeds images and sentences into the same metric space. Especially, image-caption retrieval focuses on matching the most relevant image(s) from a database with a given text query(Babenko et al., 2014). As retrieval-based system finds output candidates in a pre-structured database, it is more advantageous in overall likeliness in output qualities and shows less variation in sample mean quality. However, most of multi-modal retrieval systems deals with mapping a single instance. Few retrieval works have been explored to retrieve sequential outputs for structured queries. Some previous retrieval systems ranked images based on visual phrases(Sadeghi & Farhadi, 2011), or multi-attribute descriptions(Siddiquie et al., 2011). Kim et al. (2015) first proposed a ranking system to retrieve image sequences from natural language paragraphs. Recently, Chen et al. (2019) proposed visual segment matching framework to improve the output coherency and storyboard creation tool for how retrieval system can be applied to practical field application. Nevertheless, none of which considered the semantic arrangement with previously sampled instances.

### 2.2 SCENE GRAPHS

A scene graph depicts the contents of an image in the form of graph structure. Graph nodes represent objects, their attributes, and the relationshipo among them. As a scene graph provide visual information in abstractive level, it has been proven to be effective in a range of visual comprehension tasks such as image retrieval(Yoon et al., 2020), image or video captioning(Chen et al., 2020; Hong et al., 2020), visual question answering(Damodaran et al., 2021), and image generation(Johnson et al., 2018). A number of applications utilizing scene graph information have been widely spread after a large-scale scene graph annotations of real world images revealed from Visual Genome dataset(Krishna et al., 2017). In representation learning context, there have been many recent works focusing on learning intermediate representations of scene graphs. Those works suggest scene graph representation as an useful compressed information for downstream applications. Raboh et al. (2020) proposed differentiable scene graphs, which can be trained end-to-end with reasoning supervision. Maheshwari et al. (2021) constructed semantically rich representation through ranking loss(Karpathy & Fei-Fei, 2015; Kiros et al., 2014) coupled with triple sampling strategy in image retrieval task. The closest related work (Yoon et al., 2020) proposed experimental approach that leverages similarities of scene graph embedding for image-to-image retrieval task.

### 2.3 DUAL LEARNING

The application of dual learning was first proposed by (He et al., 2016a) to relieve the burden for preparing paired training data of English-to-French translation. The key idea of dual learning is setting a primal and a dual task in a domain translation task. Learning source-to-target(primal) and target-to-source(dual) mappings simultaneously gives ..... Especially, such a mutual reinforcing mechanism have shown effective results on generation tasks in unsupervised settings. (Luo et al., 2019)The advanced image-to-image translation GANs(Yi et al., 2017; Zhu et al., 2020) have shown competitive performance with unlabeled data by leveraging primal-dual relation to guarantee stable domain translation performance without daunting the qualities of the generated images. In our work, we set an image sequence retrieval from given story as a primal task and regenerating the original story from chosen images as a dual. The informative error signals from reconstruction(dual) task enforces the retrieval agent to choose more 'thoughtful' inputs for dual agent.

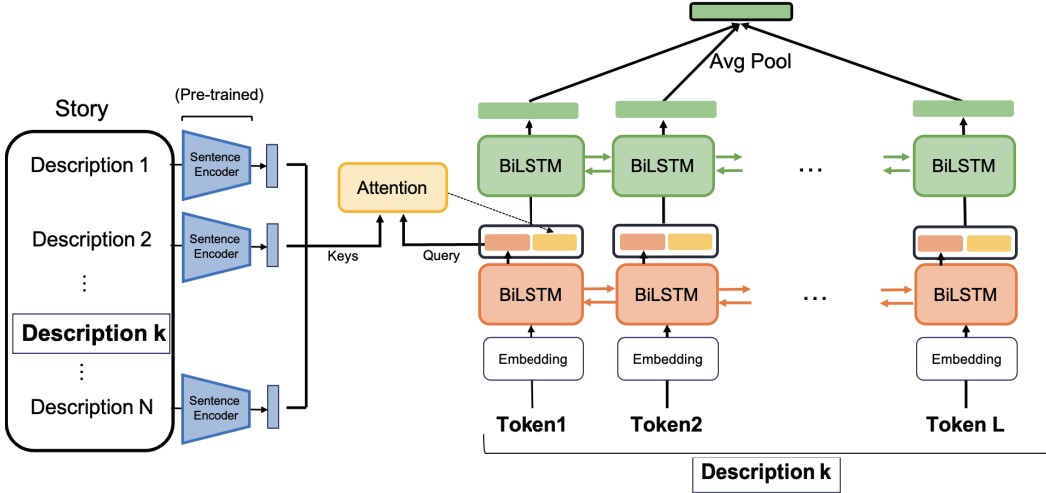

Figure 2: An illustration of overall architecture of Contextual Sentence Encoder

# 3 APPROACH

## 3.1 PROBLEM STATEMENT

Let $\mathbf{S} = \{s_i\}_{i=1}^N$ is a story comprises $N$ text descriptions, where each description composed of a single or a few sentences. For the simplicity and avoid annotation confusion, consider a single description as a single sentence. The goal of the task is retrieving an image sequence $\mathbf{I} = \{i_j\}_{j=1}^N$ that is semantically aligned to a story $S$ and each image $i_j$ is descriptive photo of the description $s_{i=j}$. Hence, the retrieval system $F : \{s_i\}_{i=1}^N \longmapsto \{i_j\}_{j=1}^j$ should map given story to the most probable image sequence without losing visual coherency. In this paper, we set $\mathbf{I}$ and $\mathbf{S}$ to have a same cardinality to easily compare the matching results by one-to-one. We left one-to-many retrieval as our future work.

## 3.2 IMAGE SEQUENCE RETRIEVAL VIA GRAPH SIMILARITIES

**Contextual Sentence Encoder.** We give a story for the direct input to our retrieval system. A story describes given image sequence(video) as a set of natural language descriptions. Many of previous text-image retrieval system receives a single text query as an input for their text encoders. If we try to process a story with traditional text encoders, contextual connection between each description will be break inevitably since there's no other way to encode a story without recurrently injecting the description. In consequence, feature representation of each description will be separately located even in text-image joint embedding space. To encourage each description to be encoded in homogeneous way, each embedding must imply contextual information of preceding text. In other words, the desired text encoder in story-to-images retrieval system must be *context-recognizable*. Thus, we need to implement a novel story encoder for the task. Inspired by Chen et al. (2019), we suggest C.S.E(Contextual Sentence Encoder) which is suitable for sequential text-to-image retrieval.

**??** describes the overall architecture of C.S.E. The key idea of C.S.E is considering structured hierarchical relation between text and it's tokens. To extract a dense representation of the description, the tokens in the description should also contain relevant information about other surrounding texts.

C.S.E efficiently encodes each description $s_i \in \mathbf{S}$, considering other surrounding descriptions in $\mathbf{S}$. C.S.E consists of bottm Bi-LSTM layer, intermediate attention layer, and final Bi-LSTM layer with global average pooling head. Let a single description in $\mathbf{S}$ as $s_i = \{w_{i_1}, w_{i_2}, ..., w_{i_L}\}$, which is a sequence of $L$ token $w_{i_t}$ s.t. $w_{i_t} \in \mathbb{R}^{|V|} 1 \le t \le L$. At time step $t$, the forward hidden state of the bottom Bi-LSTM layer $\overrightarrow{h_t^i}$ receives $e_t^i \in \mathbb{R}^{d_{emb}}$ an embedding of $e_t^i$, and the last hidden state $\overrightarrow{h_t^i}$.

The backward hidden state at time $t$ $\overleftarrow{h_t^i}$ receives arguments in the same way. The final hidden state of at time $t$ is a concatenation of $\overleftarrow{h_t^i}$ and $\overrightarrow{h_t^i}$. This can be summarized as

$$
\begin{aligned}
\overrightarrow{h_t^i} &= \overrightarrow{\mathrm{LSTM}} \left( \overrightarrow{h_{t-1}^i}, W_e w_t^i \, ; \, \overrightarrow{W_h}, \overrightarrow{b_h} \right) \\
\overleftarrow{h_t^i} &= \overleftarrow{\mathrm{LSTM}} \left( \overleftarrow{h_{t+1}^i}, W_e w_t^i \, ; \, \overleftarrow{W_h}, \overleftarrow{b_h} \right) \\
\mathrm{h}_t^i &= \left[ \overrightarrow{h_t^i} \, ; \, \overleftarrow{h_t^i} \right]
\end{aligned}
\tag{1}
$$

We can regard $h_t^i$ as a compreseed representation of token $w_{it}$. To embed the context of surrounding descriptions, we set $h_t^i$ as a query of Bahdanau attention(Bahdanau et al., 2016) module.

To compute textual coherency with the description $s_i$ which includes query $w_{it}$, we select other descriptions except $s_i$ as candidates for keys, and their hidden representations as keys. We pass all $s_k \in \mathbf{S}, 1 \leq k \leq N, s.t. k \neq i$ to pre-trained sentence encoder(Kiros et al., 2014) and extract a set of hidden representations $\{k^1, .., k^{i-1}, k^{i+1}, .., k^N\}$. Then, the value vector of attention $v_t^i$ is computed as a weighted sum of keys and queries:

$$
v_t^i = \sum_{1 \leq j \neq i \leq N} h_t^i \alpha_{ij}
\tag{2}
$$

The attention weight $\alpha_{ij}$ is a softmax score of attention layer with tanh activation computed by

$$
\begin{aligned}
\alpha_{ij} &= \frac{\exp(e_{ij})}{\sum_{k=1}^{T_x} \exp(e_{ik})} \\
e_{ij} &= v_a^\top \tanh \left( W_a h_t^i + U_a k_j \right)
\end{aligned}
\tag{3}
$$

where $v_a^\top$, $W_a$, and $U_a$ are learnable parameters of above attention layer. As $v_t^i$ is a value vector from $h_t^i$ and $\{k^1, .., k^{i-1}, k^{i+1}, .., k^N\}$, we can regard $v_t^i$ dense token representation which embeds contextual semantic relationships with other surrounding descriptions in a single story.

Now we pass the concatenation of $v_t^i$ and $h_t^i$ as a new input for the second Bi-LSTM layer. We denote the hidden state of the second RNN-based layer as $g$, and the second layer goes same progress to extract the $t^{th}$ token representation of description $s_i$ :

$$
\begin{aligned}
\overrightarrow{g_t^i} &= \overrightarrow{\mathrm{LSTM}} \left( \overrightarrow{g_{t-1}^i}, [h_t^i \, ; \, v_t^i] \, ; \, \overrightarrow{W_g}, \overrightarrow{b_g} \right) \\
\overleftarrow{g_t^i} &= \overleftarrow{\mathrm{LSTM}} \left( \overleftarrow{g_{t+1}^i}, [h_t^i \, ; \, v_t^i] \, ; \, \overleftarrow{W_g}, \overleftarrow{b_g} \right) \\
\mathrm{g}_t^i &= \left[ \overrightarrow{g_t^i} \, ; \, \overleftarrow{g_t^i} \right] \\
\mathrm{r}^i &= \mathrm{avgpool} \left( [g_1^i \, ; \, ... \, ; \, g_L^i] \right)
\end{aligned}
\tag{4}
$$

Finally, we average-pool along time steps $1 \leq t \leq L$ to get the final contextual representation of the description $s_i$.

### 3.3 Scene Graph Generation and Embedding

**Scene Graph Generation.** A scene graph is an abstract representation of the visual contents of an image(Johnson et al., 2015). Formally, we define scene graph $\mathcal{G}$ of an image $I$ as $\mathcal{G} = \{\mathcal{O}, \mathcal{R}\}$ a set of object nodes $\mathcal{O}$, and a set of relationship between two certain nodes $\mathcal{R}$. Every relationship $r_k \in \mathcal{R}$ is represented by a triplet of nodes $(subjective, predicate, objective)$, explaining dynamical association between two nodes. Predicate is represented as undirected edge. We treat GloVe(Pennington et al., 2014) embedding of the label name as a feature representation of all nodes in $\mathcal{O}$ and all edges in $\mathcal{R}$. Specific configuration will be explained in section 4. The combination of constituents of scene graph varies in related works(Chen et al., 2020; Hong et al., 2020; Yoon et al., 2020; Damodaran et al., 2021; Johnson et al., 2015; Krishna et al., 2017; Raboh et al., 2020).For example, some works(Yoon et al., 2020; Ashual & Wolf, 2019) also included a set of attributes

of objects $\mathcal{A}$ as another constituent of scene graph. In this work, we .... to relieve computational burden.

Generating scene graph $\mathcal{G}$ structure from an image $I$ is equivalent to parsing an object detection result of a target image. We used Faster R-CNN(Ren et al., 2015) as an underlying detector. For each image $I$, the detector predicts a set of region proposals $B = b_1, b_2, ..., b_n$. Each proposal $b_k \in B$ comes with bounding box feature representation and probabilities for corresponding object label. Building on these information, we applied recently proposed method(Zellers et al., 2018) for our scene graph generator. In detail, we utilized VG(Visual Genome) dataset(Krishna et al., 2017) configuration to assign proper predicate for constructing $\mathcal{R}$. Predicate label is predicted based on frequency prior knowledge from VG. In overall retrieval pipeline, we generate scene graphs from primarily picked image sequence $I$ for given story $S$ and pass to the graph encoding head layer to compare the scenery similarities via computing similarity scores of their graph embeddings.

**Encoding Scene Graphs via GCN.** In order to encode a scene graph with end-to-end fashion, we need a suitable neural architecture that can operate directly on graph-structured data. We apply *Graph Convolutional Network*(Kipf & Welling, 2016) as our main graph encoding module since it's learning ability on graph representation have been proved in many related works(Johnson et al., 2018; Yoon et al., 2020; Kipf & Welling, 2016; Chen et al., 2020).

### 3.4 Dual Learning with Video Captioning

We adopt dual learning framework for enhancing a contextual coherency of output image sequence. The primal task can be represented as $F : \mathbf{S} \longrightarrow \mathbf{I}$ with proposed retriever $F$. We denote $G$ as a dual agent, s.t. $G : \mathbf{I} \longrightarrow \mathbf{S}$. $G$ can be also regarded as visual storyteller, which generates figurative and consistent narrative for successive images. As $F$ recurrently selects $I$, we primarily considered image captioning(Karpathy & Fei-Fei, 2015) as a dual. However, since each description in a single story narrates same circumstance, reconstructed descriptions will be semantically isolated from global context. On this, the output format must be a paragraph-level captions. Hence we regard dual task as a video captioning problem, to generated semantically aligned sentences while keeping global context. We construct video captioning module based upon the proposed architecture of GLAC Net(Kim et al., 2018).

### 3.5 Training Objectives

**image Sequence Retrieval.** Let $f_{\mathcal{T}}(\,\cdot\,;\theta_{\mathcal{T}})$ a textual encoder parameterized by $\theta_{\mathcal{T}}$, which is C.S.E in this paper. $h_{\mathcal{T}}(S) = f_{\mathcal{T}}(S;\theta_{\mathcal{T}}) \in \mathbb{R}^{d_{\mathcal{T}}}$ is a dense representation of an input description $S$ which embeds contextual relations with surrounding $S$ in a story. Similarly, let $h_{\mathcal{V}}(I) = f_{\mathcal{V}}(I;\theta_{\mathcal{V}}) \in \mathbb{R}^{d_{\mathcal{V}}}$ be a feature representation from pre-trained image encoder(e.g., VGG19(Simonyan & Zisserman, 2014), ResNet152(He et al., 2016b)) before the last FC layer when given an input image $I$. We map $h_{\mathcal{T}}$ and $h_{\mathcal{V}}$ into joint embedding space through following linear transformation.

$$\begin{aligned} \phi_{\mathcal{T}}(S\,;W_{\mathcal{T}},\theta_{\mathcal{T}}) &= W_{\mathcal{T}}^{\top} h_{\mathcal{T}}(S) \in \mathbb{R}^{d_e} \\ \phi_{\mathcal{V}}(I\,;W_{\mathcal{V}},\theta_{\mathcal{V}}) &= W_{\mathcal{V}}^{\top} h_{\mathcal{V}}(I) \in \mathbb{R}^{d_e} \end{aligned} \tag{5}$$

With linear operators $W_{\mathcal{T}} \in \mathbb{R}^{d_{\mathcal{T}} \times d_e}$ and $W_{\mathcal{V}} \in \mathbb{R}^{d_{\mathcal{V}} \times d_e}$, we now can do vector computation across different modalities. We measure the similarity of two distinct modality representation through cosine-similarity base score function, defined as

$$s(s,i) = sim(\phi_{\mathcal{T}}(s), \phi_{\mathcal{V}}(i)) \tag{6}$$

where $sim(\mathbf{a}, \mathbf{b}) = \frac{\mathbf{a} \cdot \mathbf{b}}{\|\mathbf{a}\|\|\mathbf{b}\|}$. To retrieve an image which semantically matches current input description and morphologically similar with previous image, we jointly use hinge ranking losses(Faghri et al., 2017; Chechik et al., 2010; Frome et al., 2007) between correct matches and other wrong ones and graph embedding similarity. At time step $t$, the step-wise loss $\ell_t$ is

$$\begin{aligned} \ell(s,i) = &\sum_{\tilde{s}} max(\Delta - s(s,i) + s(\tilde{s},i), 0) \\ &+ \sum_{\tilde{i}} max(\Delta - s(s,i) + s(s,\tilde{i}), 0) \\ &+ 1 - s(\psi_{i_{t-1}}, \psi_{i_t}) \end{aligned}$$

with margin $\Delta$, negative description $\tilde{s}$ for $i$ and negative image sample $\tilde{i}$ for $s$. A hinge ranking loss, a.k.a triplet ranking loss, directs the retriever $F$ to minimize the first and the second term via choosing the closest counterpart $i$ or $s$ than to any unmatched samples $\tilde{i}$ or $\tilde{s}$ by margin $\Delta$. At the same time, the last term of $\ell_t$ contributes $F$ to search a photo that is structurally similar to an earlier image. The total loss for image retrieval is $L_{retrieval} = \sum_t^N \ell(s_t, i_t)$.

**Video Captioning.** After selecting aligned images as a sequence $\mathbf{I} = \{i_1, ..., i_N\}$, we input $\mathbf{I}$ and a story $S$ as a target ground-truth text, training video caption generator $G$ to reconstruct $S$. We use cross-entropy loss

$$L_{caption} = -\sum_{l=1}^{N} \sum_{v=1}^{V} y_{il}^v \log p_{il}^v \tag{7}$$

where $v \in \{1, ..., V\}$ is an index of vocabulary set. $p_{il}^v$ is predicted probability for $i - th$ token, and $y_{il}^v$ is the target token.

**Overall Objective.** The overall objective is the sum of image sequence retrieval loss and video captioning loss. The total objective is as follows:

$$L_{total} = L_{retreival} + L_{caption} \tag{8}$$

# 4 EXPERIMENTS

## 4.1 EXPERIMENTAL SETUP

**Dataset.** We evaluate the proposed method on the VIST training set for the training, and evaluate story-to-images retrieval on VIST test set. VIST includes total 210,819 unique photos within 10,117 Flickr albums. There are two data type in VIST, DII(description-in-isolation) and SIS(story-in-sequence) respectively. The DII data only contain pairs of single sentence and sinle image, while SIS contains pairs of story and images for training. For our task, we only use SIS type. A single story in SIS consists of five successive images with the corresponding captions. After excluding broken images, we finally use 40,071 stories for training, 4,998 stories for validation and 5,055 stories for test set.

**Baselines.** We compare our approach with conventional text-to-image retrieval baselines. Primarily we adopt VSE++(Faghri et al., 2017), which exploits the idea of hard negative mining(Dalal & Triggs, 2005; Felzenszwalb et al., 2009) for learning visual-semantic embeddings for cross-modal retrieval. We also apply variant of VSE++, denoted as VSE0 which uses hinge-based triplet ranking loss(Karpathy & Fei-Fei, 2015; Kiros et al., 2014; Socher et al., 2014). Since both of them are single entry retrieval model, we also compare our retrieval system with existing sequential text-to-image model, CNSI(Ravi et al., 2018). CNSI is a global visual semantic matching model that utilizes pre-computed modality feature as an encoder. Lastly, we conduct ablation study to examine the power of contextual text encoding, which is implemented by comparing our suggested retriever and the retriver without CSE.

**Metrics.** We use $Recall@K(k = \{1, 5, 10\})$ for main evaluation metric for VIST. For each description in the story, we retrieve top-K image predictions and measure the total percentage of sentence descriptions whose ground-truth images are whether ranked in the top-K predictions. Hence, the desired retrieval system maximizes recall at top-K. Also, we jointly evaluate on common retrieval metrics including median rank (MedR).

**Hyperparameters.** The target parameters are included in CSE, graph encoding head, and video captioning module. We unified optimizers for each module with Adam, setting distinct initial learning rates. In order, we set 0.0002, 0.0001, and 0.001. We only decay the learning rate for CSE, keeping initial learning rate for the first 15 epochs and then lower the learning rate to 0.00002 for remaining epochs. We set a minibatch size as 32, all parameters are trained for 30 epochs. We employ 300-dimensional GloVe as a feature representations of nodes and relations in a scene graph.

## 4.2 QUANTITATIVE RESULTS

Table 1 shows story-to-images retrieval performance on the VIST testing set. Overall, We observe sequential retrieval system(CNSI, Ours) performs better than single entry retrieval models(VSE0,

Table 1: story-to-images retrieval performance on the VIST testing set. All scores are reported in percentage(%) .

| Method | R@1 | R@5 | R@10 | Med r |
|---|---|---|---|---|
| VSE 0 | 11.25 | 12.27 | 12.31 | 11.74 |
| VSE++ | 12.28 | 12.29 | 13.27 | 12.57 |
| CNSI | 13.01 | 13.99 | 14.27 | 13.77 |
| Ours(w/o CSE) | 10.39 | 11.48 | 12.10 | 11.50 |
| **Ours** | **13.35** | **14.07** | **14.40** | **13.98** |

VSE++). In story-to-images setting, we could easily expect this kind of result since the formers(CNSI, Ours) sequentially embeds features from a given story compared to the latters(VSE0, VSE++). Besides, we presume usage of hard negatives for objective function for retrieval can bring positive effect for increasing the retrieval performance through comparing the results of VSE series. CNSI yields the best performance among baselines. Because CSE is one of the main contributors to maintain global semantic context in a story, our suggested retriever without CSE shows comparable performance with VSE series. Overall, our suggested pipeline outperforms the baselines. We empirically observe that leveraging scene graph similarity and dual framework helps gives better predictions in retrieved images. Nevertheless, there are not dramatic increases in performances. We assume that even the design choices of architecture for the main system pretty differs a lot, the differences in recall percentages of top-K output samples are relatively trivial. We leave evaluation on larger K, and other visual storytelling benchmark for our future work.

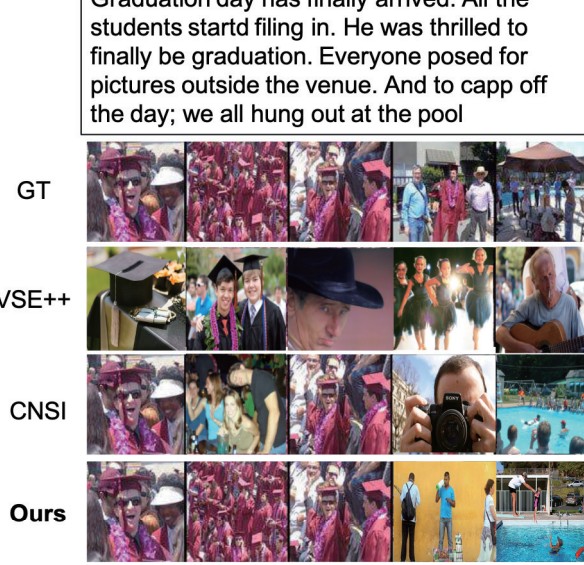

Figure 3: Qualitative comparison on predicted images from test set sample.

## 4.3 QUALITATIVE ANALYSIS

Figure 3 depicts samples of predicted images for when given a random story in the test set. We can observe that VSE++ does not maintain global context in both images and text. CNSI shows better result, still less incoherent. Compared to others, our retrieval system provides better visual descriptions. Nevertheless, we conclude there's a lot of room to develop the performance in qualitative way.

## 5 CONCLUSION

In this paper, we introduced a new story-to-images retrieval framework that can alleviate potential pitfall of sampling visually incoherent images from a database. Our main technical contributions include (1) utilizing scene graph similarities with prior sample and (2)apply video captioning as a dual task. Our suggested framework shows superior performance in VIST benchmark compared to conventional text-to-image retrieval works.

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
