# OpenReview forum: "Graph Similarities and Dual Approach for Sequential Text-to-Image Retrieval"
_ICLR.cc/2022/Conference — ICLR 2022 Submitted_

### Official Review · Reviewer_iXYw · 2021-11-02

**Correctness:** 4
**Technical Novelty And Significance:** 3
**Empirical Novelty And Significance:** 2
**Recommendation:** 5
**Confidence:** 3

**Main Review:**

Strengths:
- For the some part, the paper is well written and easy to read, especially the 1st 2 sections, I felt that the motivations are strong.
- The proposed system cleverly leverages existing techniques (CSE, graph generation & embedding, dual learning) to realize the motivations: CSE produces context informative embedding of text, and the scene graph based image retrieval helps improving global contextual consistency of retrieved images. The dual learning task is basically the reverse translation task makes sense; it is commonly seen in the literature, helping the forward translation task more consistent.

Weaknesses:
I have the following troubles understanding the proposed system:
- (1) The images retriever part is figure 1 is not clear (or was not explained at all). It seems that it performs retrieval of 1 image at a time; at each iteration, there is a 1st retrieval phase and a 2nd retrieval phase? I wasn't able to piece them together by just looking at the figure.
- (2) 1 of the motivation of the proposed is that existing works "neglect scenery coherence among images", I was expecting that something in section 3 would explicitly address that? Section 3.3 is just describing some existing work that is being used here, not connecting everything together, or maybe the authors have not finish writing this section?
- (3) It's not clear what is the goal of the retrieval loss in section 3.5 where VGG/ResNet is being used as image encoder. I'm guessing they are used to train the CSE only, and the pretrained VGG/ResNet is never used in testing? But so which loss optimizes the graph generation and embedding part of the system?

I feel that the experiments are not extensive enough
- (4) no ablation study, this is the biggest weakness of this work I think. It is good to study & validate each components effect, whether they are leveraged from existing works or newly proposed. Have we tried a different text embeder, or graph generation method, or graph embedding method, or ranking loss, etc. What is the effect of the additional dual learning task? I think the ours vs ours w/o CSE is very insightful comparison, but more is needed, it's not enough to just say "We empirically observe that leveraging scene graph similarity and dual framework helps gives better predictions in retrieved images" without convincing evidence & extensive study.
- (5) While it is expected that the proposed would outperform the single-image-to-text methods (VSE0, VSE++), the equivalent proposed (ours w/o CSE) really under perform making me think that maybe the "backbone" of the proposed is not well optimized. That might be because of the lack of validations mentioned above.
- (6) Maybe more qualitative analysis, depict both cases where it does well and cases where it doesn't.

typos: ?? in page 4, last paragraph in page 3, first line of page 6,

**Summary Of The Paper:**

This paper proposes a new story to images retrieval system (or sequence-of-texts to sequence-of-images retrieval).
It consist of: (1) the Contextual Sentence Encoder (CSE) that embeds description text in each story taking advantage of the context, that is each description embedding feature also absorbing information from other description in the same story, (2) the scene graph generation and embedding module to encode visual contents of images, and (3) a dual learning task that tries to regenerate the original story from image sequence retrieval result.
The proposed system outperforms other method on the VIST testset.

**Summary Of The Review:**

This work is interesting but underdeveloped.

I think the proposed system passes the bar as a novel method. While most of its component are from existing works, the integration and design of the whole system for the story-to-images retrieval task is a significant technical contribution.

However, the experiment & ablation study are not extensive enough. Writings seem unfinished last few sections of the paper.

So even though this work is very promising, I'd recommend not accept the paper at the current state. The authors are encouraged to spend more time polishing it.

---

### Official Review · Reviewer_6mhj · 2021-11-02

**Correctness:** 2
**Technical Novelty And Significance:** 2
**Empirical Novelty And Significance:** 2
**Recommendation:** 3
**Confidence:** 5

**Main Review:**

Strength:

+ The idea of optimizing graph embedding similarities among sampled images is interesting.

+ The proposed method shows some strength compared with some existing methods.

+ The paper is easy to follow.


Concerns:

- The comparisons are actually unfair with existing methods. Data and annotations related to scene graph generation (e.g. Visual genome dataset) are used for training the proposed model. However, existing methods do not rely on such data and labels. Comparisons in Table 1 is unfair.

- Based on unfair comparisons, the strength of the proposed model is quite limited. It achieves only 0.34 % improvements compared with CNSI which is published in year 2018. VSE++ and VSE0 are very basic baseline models in the common image-text matching setting. Why not including more recent image-text matching models rather than VSE++ into comparisons?

- Experimental designs and results are not convincing. There is no ablation studies to show and evaluate the contributions of each part of the model. The complex designs here may be redundant.

- Technical contributions are also limited. Using scene graph for cross-modal retrieval is not new [a]. Considering both retrieval and generation is also not new [b]. Considering both graph space and joint space is interesting but not novel enough and its effectiveness is not well supported due to the missing ablation studies.

[a] Cross-modal Scene Graph Matching for Relationship-aware Image-Text Retrieval, WACV 2020
[b] Look, Imagine and Match: Improving Textual-Visual Cross-Modal Retrieval with Generative Models, CVPR 2018

- The writing needs improvements and many typos need to be revised, e.g. Eq, L_retreival -> L_retrieval,  image Sequence Retrieval -> Image Sequence Retrieval, etc (not listing all of them).


**Summary Of The Paper:**

The paper focuses on the topic of sequential text-to-image retrieval, where the goal is to match text sequences and images. Specifically, authors propose to use scene graph representation of images and learn to maximize graph embedding similarities among sampled images, so that morphological relevance with previously sampled images is considered during the process. Dual learning task is also constructed by calculating a video captioning loss.

Experiments are conducted on VIST dataset as for story-to-images retrieval, which shows the strength of the proposed method compared with some baseline models. It performs around 0.34 % better than works published before year 2018.

**Summary Of The Review:**

The review is mainly based on the technical novelties/contributions, experimental designs, state-of-the-art comparisons, etc. Please refer to the Strength and Concerns parts in the Main Review section for more details.

---

### Official Review · Reviewer_1pAQ · 2021-11-04

**Correctness:** 2
**Technical Novelty And Significance:** 1
**Empirical Novelty And Significance:** 2
**Recommendation:** 1
**Confidence:** 5

**Main Review:**

# Strengths
The authors are on the right track with the method design. It is clear that capturing the context of other sentences (i.e. baking the context of the entire story into a single sentence embedding) is important to ensure that the retrieved image is not just feasible for the given sentence, but also for possible future sentences. The authors try to address this by using a contextualized embedding method for each sentence.
Additionally, the authors identify that visual stories should also usually have some visual consistency. In the example they give with a ball being kicked on the roof, the authors show one image with a ball in the air, next it is moving higher, and eventually we see an image of a house with the ball on the roof. Thus, the authors identify that their should be some sort of continuity between the scene graphs for each textual description (as image_{t-1} shifts to image_{t}). Thus they extract scenes and impose a constraint that the retrieved image shouldn't be too far from the previously retrieved scene graph.

I also like the idea of using cycle consistency (that is to say, once we retrieve a set of images from a textual story, can we try to regenerate that same story from the set of images). However, I have a lot of questions about that that I will discuss in the weaknesses section, however I think the basic premise is reasonable.

# Weaknesses
Unfortunately, the paper suffers from major weaknesses. Before getting into technicalities, the paper draft seems highly unpolished, draftlike, and incomplete. For example, Figure 1 is impossible to read and had to be zoomed to an extremely high degree to understand. Other portions of the paper are incomplete. For example, on page 3, sentences just trail off. "Learning source-to-target(primal) and target-to-source(dual) mappings simultaneously gives......"
There are numerous occasions throughout the text where there are "placeholder" ellipses. Also, the top of page 6 ("In this work, we ......").
There are spelling errors throughout the text (e.g. "compreseed"), inconsistent capitalization of sections (e.g. page 6- image Sequence Retrieval is lower case "i"), and the text is otherwise poorly written, hard to follow, and in many instances too light on details to adequately understand what is being done.

I will briefly mention technical critiques now. First, the authors method of using "contextual sentence encoder" is as far as I can tell essentially identical to the Chen et al. 2019 reference. Thus, there is no novelty on the "CSE" portion of this. The "graph encoding" portion is very light on details, but also appears to primarily rely on off-the-shelf methods methods (e.g. Faster R-CNN, other cited works - e.g. Zellers, et al.). There are no details about the GCN used, but it seems to be a standard model and I can't assess any novelty there.
The authors use a standard triplet loss to train the method, with an extra constraint that imposes that the retrieved image's graph embedding should be close to the prior one.
One shortcoming of this is that it isn't conditioned on the text at all. For example, there are many visual stories where it makes sense that each image should be consistent with its prior image. But what if there is a sudden turn in the story? What if the text says something about a different scene or location (this occurs in the pool party example in the single qualitative result). In this case, the model will be constrained to pick an image with a similar graph embedding as a previous image, but this may actually not be desirable in that case. A better strategy would be to weight the graph consistency based on some relationship of text query $t$ with $t-1$.

I am also very concerned about the "captioning" consistency piece. The authors mention this as a critical part of their method, but there is only an inch of space devoted to it on page 7. Basically, the idea is after we select images, we want to use a video captioning method to regenerate the original story. This is a good idea in practice, but I am very concerned about this. First of all, the image retrieval method retrieves an image at each time step. The authors then say, after selecting the aligned images, they input them into a video captioning method and train it with a loss on the original story text (ground truth text). Ideally, what we would want to do is backpropagate through this captioning method, back through the image retrieval method, and through the entire system.
However, I'm not sure this is possible. How can the gradients of the video captioner be backpropped back through the retrieval process? Analogous to image captioning methods, once one generates words from a captioning method, one cannot then backpropagate through the chosen words (since the word selection is non-differentiable). Here, since there is a set of images, and presumably the authors take top-1, how does the captioning method get to consider other possibilities? How is this process differentiable? It would seem some sort of reinforcement learning is necessary - but there are no details.

For experiments, the authors compare against older baselines (e.g. VSE++), when much more recent methods are available. The experiments are very underwhelming. We see the authors method for R@1 gets 13.35, while best baseline gets 13.01. Median r of authors' method is 13.98, while best baseline is 13.77. In sum, there is no significant gain over prior state of the art work. I would also argue that the authors need to do a much more rigorous experimental comparison to more related work (e.g. Batra, Vishwash, et al. "Variational Recurrent Sequence-to-Sequence Retrieval for Stepwise Illustration." Advances in Information Retrieval 12035 (2020): 50.). The authors also include a single qualitative result, but it isn't clear how representative it is or that the results are even particularly meaningful. No human study is done to assess the quality of the chosen "stories".

**Summary Of The Paper:**

The authors propose a method for a sequence-to-sequence retrieval task. The idea of the task is given a set of sentences describing steps in a coherent story, the model needs to retrieve a single image per each sentence to illustrate that story. Unlike standard cross-modal retrieval where the goal is to just retrieve the image from a single caption, ignoring context, here the goal is to retrieve an image that is consistent with the retrieved visual story so far (i.e. each step in the story should be visually consistent across the retrieved images).

The authors method is designed to address these unique aspects of the problem. First, the authors encode a given sentence from the story using a contextual embedding computed using surrounding captions. They use this contextual sentence embedding to perform cross-modal retrieval of images. After doing this, they also take the previously chosen image and extract its scene graph. They essentially then compute a distance using both the scene graph embedding (from prior retrieved images) and the sentence embedding to determine the image at that time step. Finally, the authors take the set of retrieved images to generate captions and then measure the consistency of the generated captions with the story.

The authors present preliminary experimental results showing that the proposed method has very slight gains on the VIST (Visual Storytelling) dataset. They also present a qualitative result showing a retrieved "visual story."

**Summary Of The Review:**

In sum, the authors tackle the problem of visual storytelling by proposing a method that captures sentence context and prior chosen image scene structure for sequence to sequence retrieval. The task is interesting, however, the authors' method lacks technical novelty and key components of the method are unclear. The paper is highly draft-like, with figures unreadable, portions unwritten, and writing errors throughout. The experiments show slight gains of the authors' method, but it isn't clear if these gains are significant and there is a lack of comparison to recent relevant research.
At present, the paper is not ready for publication and needs substantial work.

---

> ### Comment · Reviewer_1pAQ · 2021-11-30
> **final score**
>
> I have reviewed all other reviews and keep my original score.

---

### Decision · Program_Chairs · 2022-01-20

**Decision:**

Reject

**Comment:**

Reviewers unanimously vote for rejection for several reasons. First, the draft is incomplete and difficult to read. Second, one of the proposed methods (contextual sentence encoder) appears the same as past work, while the other proposed method (graph encoding) is difficult to interpret from what is written. Third, the draft is missing comparisons with recent work, and some included comparisons may be unfair due to data conditions. No author response was provided. The reviewer consensus is that this draft is underdeveloped, and not yet ready for submission or publication.